# Bruton’s Tyrosine Kinase Inhibitor Zanubrutinib Effectively Modulates Cancer Resistance by Inhibiting Anthracycline Metabolism and Efflux

**DOI:** 10.3390/pharmaceutics14101994

**Published:** 2022-09-21

**Authors:** Lucie Čermáková, Jakub Hofman, Lenka Laštovičková, Lucie Havlíčková, Ivona Špringrová, Eva Novotná, Vladimír Wsól

**Affiliations:** 1Department of Biochemical Sciences, Faculty of Pharmacy, Charles University, Akademika Heyrovskeho 1203, 50005 Hradec Kralove, Czech Republic; 2Department of Pharmacology and Toxicology, Faculty of Pharmacy, Charles University, Akademika Heyrovskeho 1203, 50005 Hradec Kralove, Czech Republic

**Keywords:** zanubrutinib, anthracycline, drug resistance, aldo-keto reductase 1C3, ABC drug efflux transporter

## Abstract

Zanubrutinib (ZAN) is a Bruton’s tyrosine kinase inhibitor recently approved for the treatment of some non-Hodgkin lymphomas. In clinical trials, ZAN is often combined with standard anthracycline (ANT) chemotherapy. Although ANTs are generally effective, drug resistance is a crucial obstacle that leads to treatment discontinuation. This study showed that ZAN counteracts ANT resistance by targeting aldo-keto reductase 1C3 (AKR1C3) and ATP-binding cassette (ABC) transporters. AKR1C3 catalyses the transformation of ANTs to less potent hydroxy-metabolites, whereas transporters decrease the ANT-effective concentrations by pumping them out of the cancer cells. In our experiments, ZAN inhibited the AKR1C3-mediated inactivation of daunorubicin (DAUN) at both the recombinant and cellular levels. In the drug combination experiments, ZAN synergistically sensitised AKR1C3-expressing HCT116 and A549 cells to DAUN treatment. Gene induction studies further confirmed that ZAN did not increase the intracellular level of AKR1C3 mRNA; thus, the drug combination effect is not abolished by enzyme induction. Finally, in accumulation assays, ZAN was found to interfere with the DAUN efflux mediated by the ABCB1, ABCG2, and ABCC1 transporters, which might further contribute to the reversal of ANT resistance. In summary, our data provide the rationale for ZAN inclusion in ANT-based therapy and suggest its potential for the treatment of tumours expressing AKR1C3 and/or the above-mentioned ABC transporters.

## 1. Introduction

In recent decades, an understanding of the processes related to the pathogenesis of cancer has given rise to targeted therapy. Anticancer drugs target a wide range of cellular structures and signalling pathway components involved in cancer cell growth and survival. One such important target is Bruton’s tyrosine kinase (BTK), belonging to non-receptor tyrosine kinases of the TEC family [1,2]. BTK has been recognised as a key player in the B-cell receptor signalling pathway, the dysregulation of which is implicated in B-cell malignancies [3,4]. Abundant expression of BTK has been identified in B-cell chronic lymphocytic leukaemia (CLL) and B-cell lymphomas, making it a promising target for the development of new small-molecule inhibitors [5,6,7].

The first BTK inhibitor that was given accelerated approval by the U.S. Food and Drug Administration (FDA) was ibrutinib (IBR) in 2013, followed by acalabrutinib (ACA) in 2017 [8,9]. The incorporation of IBR into cancer therapy is a milestone that has changed the fate of many patients with CLL, Waldenström’s macroglobulinaemia, and some non-Hodgkin lymphomas [10,11,12,13,14]. IBR and ACA are strong BTK inhibitors that covalently and irreversibly interact with Cys-481 in the ATP-binding domain of BTK. The inhibitory activity of IBR has also been demonstrated in other kinases containing homologous cysteine residues such as Itk and Tec [6,15,16]. The combined inhibition of different kinases may contribute to some adverse effects of IBR, such as bleeding and cardiovascular complications [17,18]. Thus, although IBR and ACA are the most effective agents for treating several B-cell malignancies, treatment-limiting adverse effects and emerging cases of cancer resistance [19,20] have encouraged the search for new BTK inhibitors. Various compounds (e.g., vecabrutinib, fenebrutinib, and tirabrutinib) have been identified as BTK inhibitors and have entered clinical trials in recent years [21,22]. However, in addition to IBR and ACA, only zanubrutinib (ZAN) has been approved by the FDA [23,24,25].

In clinical trials, IBR, ACA, and ZAN have been tested alone and in combination with anthracycline (ANT)-containing standard chemotherapy [26,27,28,29]. ANTs, even in the era of targeted therapy, represent the gold standard for the treatment of haematological malignancies. Daunorubicin (DAUN) and doxorubicin (DOX) are routinely used to treat acute myeloid leukaemia (AML), acute lymphoblastic leukaemia, and aggressive lymphomas. Therefore, emerging ANT resistance is a serious complication that is often responsible for the failure of cancer therapy. The decreased cancer sensitivity to ANTs may be due to several mechanisms. Metabolic drug inactivation by anthracycline reductases and increased efflux mediated by ATP-binding cassette (ABC) transporters are known to contribute significantly to this phenomenon [30,31,32,33].

Recently, we reported that IBR and ACA interact with aldo-keto reductase 1C3 (AKR1C3) [34], an enzyme that is overexpressed in several malignancies [35,36,37,38]. AKR1C3 catalyses the two-electron reduction of DAUN and DOX, converting them to alcohol metabolites that possess lower anticancer activities than their parent drugs [32,39,40]. In our previous experiments, IBR and ACA inhibited the AKR1C3-mediated inactivation of DAUN, leading to an increase in its effectiveness against cancer cells [34]. Furthermore, IBR sensitises cancer cells to DOX by inhibiting the ABCC1 transporter [41]. Knowing these off-target activities of IBR and ACA and considering their structural similarities within BTK inhibitors, we aimed to investigate whether there might be a rationale for combining the novel second-generation BTK inhibitor ZAN and ANTs for therapeutic purposes. The assumption that targeting AKR1C3 and/or ABC transporters may improve the clinical outcome of ANTs [42,43] allowed us to investigate whether ZAN interacts with these resistance mediators and counteracts their activities. First, we performed inhibitory studies on recombinant enzymes at the cellular level. Next, drug combination studies were performed to examine whether ZAN acts as a dual-activity resistance modulator. As AKR1C3 may influence the resistance phenotype of cancer cells, we performed induction studies focusing on the expression of this enzyme. 

## 2. Materials and Methods

### 2.1. Material

ZAN was purchased from MedChemExpress LLC (Monmouth Junction, NJ, USA). Nicotinamide adenine dinucleotide phosphate (NADP+), glucose-6-phosphate, glucose-6-phosphate dehydrogenase (Roche), fetal bovine serum (FBS), and high-performance liquid chromatography (HPLC) grade solvents were obtained from Merck (Prague, Czech Republic). Phenazine methosulfate (PMS) and 2,3-Bis-(2-Methoxy-4-Nitro-5-Sulfophenyl)-2H-Tetrazolium-5-Carboxanilide (XTT) tetrazolium salt were purchased from BioTech (Prague, Czech Republic). DAUN was supplied by SelleckChem (Houston, TX, USA). Daunorubicinol (DAUN-OL), a C13-hydroxy-metabolite of DAUN, was obtained from Toronto Research Chemicals (Toronto, ON, Canada). JetPrime transfection reagent (Polyplus Transfection, Illkirch, France) was provided by VWR International Ltd. (Radnor, PA, USA). Cell culture reagents were obtained from Lonza (Walkersville, MD, USA) and Merck (St. Louis, MO, USA). TRI reagent solution was purchased from Molecular Research Center Inc. (Cincinnati, OH, USA). The RNA extraction kit was supplied by Zymo Research (Irvine, CA, USA). AKR1C3-specific primers (Forward primer: 5′-GCCAGGTGAGGAACTTTCAC-3′, Reverse primer: 5′-AGTTTGACACCCCAATGGAC-3′) and oligo-dT were purchased from Generi Biotech (Hradec Králové, Czech Republic). ProtoScript II buffer, dithiothreitol (DTT), dNTP mix, and ProtoScript II Reverse Transcriptase were obtained from New England BioLabs (Ipswich, MA, USA). The qPCR SG mix was acquired from the Institute of Applied Biotechnologies (Prague, Czech Republic). 

### 2.2. Cell Culture

Human colorectal carcinoma HCT116 cell line and human bone marrow AML KG1α cells were purchased from the European Collection of Authenticated Cell Cultures (Salisbury, UK). Human liver carcinoma HepG2 and human non-small cell lung carcinoma A549 cells were obtained from the American Type Culture Collection (Manassas, VA, USA). HCT116, HepG2, and A549 cells were cultured in Dulbecco’s modified Eagle’s medium (DMEM) with 10% FBS. KG1α leukaemia cells were maintained in Iscove’s modified Dulbecco’s medium (IDMEM) supplemented with 20% FBS and 2 mM L-glutamine. All cells were cultured under standard conditions (37 °C, 5% CO_2_) and examined for mycoplasma infections. Routine cultivation and all experiments were performed in an antibiotic-free medium. Cell passages of 10–25 were used in the experiments. ZAN was dissolved in dimethyl sulfoxide (DMSO), and the concentration of DMSO in the experiments did not exceed 0.5% (*v/v*). The effect of DMSO on cell viability was observed to be negligible in the validation experiments. Furthermore, possible interfering effects on other biological processes were eliminated by using of vehicle controls.

### 2.3. Cloning, Overexpression, and Purification of Recombinant Enzymes

As described previously, human recombinant AKR1C3 was prepared in *E. coli* BL21(DE3) [44] and purified using NGC chromatography system equipped with a 1 ml HisTrap FF column (GE Healthcare Life Sciences, Marlborough, MA, USA) [45].

### 2.4. Inhibitory Assays

To determine the value of half-maximal inhibitory concentration (IC_50_), human recombinant AKR1C3 (1.5 µg) was incubated with ZAN (0.01–100 µM) or vehicle DMSO. The reaction was carried out in 0.1 M sodium phosphate buffer (pH 7.4) containing NADP+ regeneration system (2.54 mM NADP, 19.72 mM glucose-6-phosphate, 10 mM MgCl_2_, and 0.88 U glucose-6-phosphate-dehydrogenase), as described previously [45]. After 10 min of pre-incubation, the reactions were started with a DAUN (500 µM) addition, and the solution was further incubated for 30 min. Methanol (300 µL) was used to stop the reactions, and the samples were cooled on ice and allowed to precipitate for 1 h at 4 °C. After centrifugation (9050× *g*, 10 min), the solution was filtered through a 0.2 µm PTFE membrane (Whatman, GE Healthcare, Uppsala, Sweden). The amount of DAUN-OL was analysed using ultra-high-performance liquid chromatography (UHPLC). To determine the mode of interaction between AKR1C3 and ZAN, AKR1C3 was incubated with DAUN (200–2000 µM) and ZAN (1, 5, 10 μM) or DMSO, and the Michaelis–Menten kinetic parameters were calculated and transformed into a Lineweaver–Burk plot using GraphPad Prism 9.3.1 (GraphPad Software, Inc., La Jolla, CA, USA).

### 2.5. Inhibition of AKR1C3 in Transiently Transfected HCT116 Cells

Vector coding for AKR1C3 (pCI_AKR1C3) and pCI empty vector (EV) was generated, purified from *E. coli* HB101, and used to transfect HCT116 cells, as described previously [43,46]. In brief, HCT116 cells (approximately 30 × 10^4^ cells/well) were seeded into 24-well plates and cultured for 24 h under standard conditions (5% CO_2_, 37 °C). The transfection mixture (0.25 µg plasmid and 0.75 µL jetPRIME^®^ Transfection Reagent in jetPRIME buffer) was prepared according to the manufacturer’s protocol (Polyplus Transfection, Illkirch, France). After 24 h, the culture medium was refreshed, the transfection mixture (37.5 µL) was added dropwise to each well, and the cells were incubated (37 °C, 5% CO_2_) for 24 h before being used for subsequent experiments. The functional expression of AKR1C3 was verified, as previously described [43]. To study the inhibition of AKR1C3 in HCT116-transfected cells, the culture medium was harvested and replaced with fresh medium containing DAUN (1 µM) and ZAN (1, 5, 10, and 50 µM) or DMSO (vehicle). HCT116 cells harbouring EV (HCT116-EV) or pCI_AKR1C3 (HCT116-AKR1C3) were incubated with DAUN and ZAN, or its vehicle DMSO, for 2 and 4 h (37 °C, 5% CO_2_). At specified intervals, the culture medium was collected and cells were lysed with 200 µL of lysis buffer (25 mM Tris, 150 mM NaCl, and 1% Triton X-100, pH 7.8) for 15 min at room temperature (25 °C). The medium was mixed with the cell lysate, DAUN and DAUN-OL were extracted twice with 1 mL of ethyl acetate, and the organic phases were evaporated under vacuum. These residues were then dissolved in the mobile phase and analysed using UHPLC.

### 2.6. UHPLC

The amount of DAUN-OL was determined using UHPLC with fluorescence detection. Briefly, an Agilent 1290 Series UHPLC chromatographic system equipped with a Zorbax C18 Eclipse Plus (2.1 × 50 mm, 1.8 µm) column and a 1290 Infinity inline filter (Agilent, Santa Clara, CA, USA) was used. The mobile phase was a mixture of 0.1% formic acid in water and acetonitrile at a ratio of 74:26 (*v/v*). Isocratic elution and a flow rate of 0.7 mL/min were used. The excitation and emission wavelengths of the detector were 480 and 560 nm, respectively.

### 2.7. Drug Combination Assays

Prior to drug combination studies, HCT116 cells were transfected using the same procedure as described above, but optimised for the 96-well plate. In detail, HCT116 cells (approximately 8 × 10^3^ cells/well) were seeded in 96-well plates and cultured for 24 h (5% CO_2_, 37 °C). The next day, the cells were transfected with pCI_AKR1C3 or EV, according to the manufacturer’s protocol (Polyplus Transfection, Illkirch, France). In each well of the 96-well plate, a mixture of 0.10 µg plasmid and 0.20 µL jetPRIME^®^ Transfection Reagent in jetPRIME buffer was added and incubated for 10 min at room temperature (25 °C). The polyplexes (5 µL) were then added dropwise to HCT116 cells cultured in 100 µL of fresh DMEM supplemented with 10% FBS and further incubated for 24 h (5% CO_2_, 37 °C). A549 cells (approximately 5 × 10^3^ cells/well) were seeded into 96-well plates, cultured under standard conditions (37 °C, 5% CO_2_) for 24 h, and then directly used for the experiments. To determine whether ZAN counteracts AKR1C3-mediated DAUN resistance, the culture medium was changed to a fresh medium containing DAUN (0.1–5 µM for HCT116-AKR1C3 and HCT116-EV; 0.01–1 µM for A549) and ZAN (5 and 10 µM for HCT116-AKR1C3 and HCT116-EV; 1 and 5 µM for A549), or DMSO (a vehicle). The cells were incubated for 72 h under standard conditions (37 °C, 5% CO_2_) before their viability assessment by XTT assay. Apart from these combinations, the cytotoxicity of ZAN itself (0.1–50 µM) was also determined for both cell lines. The Chou–Talalay method was used to quantify combination effects [47,48].

### 2.8. XTT Cell Proliferation Assay

An XTT solution consisting of XTT powder (1 mg/mL) and phenazine methosulfate (3 mg/mL) in phosphate-buffered saline (PBS) was added to HCT116, A549, HepG2, and KG1α cells and incubated for 1.5 h (HCT116, A549, HepG2) or 4 h (KG1α). Absorbance was measured at 450 nm using a microplate reader (Infinite M200; Tecan, Salzburg, Austria). This method was used to verify the lack of ZAN toxicity in cellular inhibitory assays, drug combination studies, and induction experiments.

### 2.9. Accumulation Studies in KG1α and A549 Cells

To study the effect of ZAN on DAUN accumulation, KG1α (approximately 50 × 10^4^ cells/well) and A549 cells (approximately 35 × 10^4^ cells/well) were seeded into 24- and 12-well plates, respectively. After 24 h, ZAN (1, 5, 10, and 25 μM), LY335979 (2 μM), Ko143 (2 μM), and MK571 (25 μM) (selective inhibitors of the ABC transporters ABCB1, ABCG2, and ABCC1, respectively) were added to the cells and incubated for 15 min. Then, 2 and 1 µM DAUN were pipetted into A549 and KG1α cell cultures, respectively. Following 1 h of incubation, the cells were washed twice with cold PBS and used for further analysis of DAUN accumulation using a Sony SA3800 Spectral Cell Analyser (Sony Biotechnology, San Jose, CA, USA) flow cytometer with a laser set to 488 nm. The SA3800 software (Sony Biotechnology, San Jose, CA, USA) was used to evaluate the data.

### 2.10. Determination of ZAN Cytotoxicity in HepG2 and KG1α Cells

Prior to the induction studies, the cytotoxicity of ZAN alone was tested to determine a relatively non-toxic concentration suitable for subsequent experiments. HepG2 (approximately 18 × 10^3^ cells/well) and KG1α (approximately 25 × 10^3^ cells/well) cells were seeded into 96-well plates, and either on the same day (KG1α), or after a 24 h-incubation (HepG2 cells), were treated with ZAN (0.1–50 µM) or DMSO (vehicle). After incubation for 48 h under standard conditions (37 °C, 5% CO_2_), the effect of ZAN on cancer cell viability was measured using the XTT assay.

### 2.11. Induction Studies in HepG2 and KG1α Cells

Induction studies were performed as previously described [46]. HepG2 (approximately 20 × 10^4^ cells/well) and KG1α (approximately 35 × 10^4^ cells/well) cells were seeded into 48-well plates. While the KG1α cells were immediately treated with ZAN (0.5 µM) or DMSO (a vehicle), the HepG2 cells were first cultured for 24 h and then treated with 0.5 µM ZAN or a vehicle control. After 24 and 48 h of incubation, the medium was removed, the cells were lysed in TriReagent, and the total RNA was isolated using the Zymo Research Direct-zol^TM^ RNA Miniprep kit and transcribed into cDNA, according to the manufacturer’s instructions (ProtoScript^®^ II Reverse Transcriptase, New England Biolabs, Ipswich, MA, USA). AKR1C3 mRNA levels were quantified using quantitative real-time PCR (qRT-PCR). The PCR contained 1x concentrated XCEED qPCR SG Mix, AKR1C3-specific primers (1 μM), and 20 ng of cDNA. qPCR was performed using QuantStudio 6flex (Applied Biosystems by Life Technologies, Carlsbad, CA, USA) set to the following conditions: initial denaturation at 95 °C for 10 min, followed by 40 cycles at 95 °C for 15 s and 65 °C for 1 min. Absolute quantification of AKR1C3 expression was achieved by comparing samples with the concomitantly amplified AKR1C3 standard, which was generated as described previously [43].

### 2.12. Statistical Analysis

Statistical analysis of the data was performed using GraphPad Prism software version 9.3.1 (GraphPad Software Inc., La Jolla, CA, USA) using ordinary one-way analysis of variance (ANOVA) followed by Dunnett’s post hoc test or unpaired t-test with Welch’s correction, as specified in the respective figure legends. Statistical significance was set at *p* < 0.05. In the case of drug combination studies, the combination indices (CI) were used to distinguish between synergism (CI < 0.9), additivity (CI = 0.9–1.1), and antagonism (CI > 1.1), which were calculated using CompuSyn software (version 1.0; ComboSyn Inc., Paramus, NJ, USA). 

## 3. Results

### 3.1. ZAN Inhibits the Recombinant AKR1C3-Mediated DAUN Reduction In Vitro

First, we investigated whether ZAN could inhibit ANT metabolism using the recombinant enzyme. AKR1C3 was incubated with the substrate DAUN, with or without ZAN. As can be seen from the half-maximal inhibitory concentration (IC_50_) (Figure 1A), ZAN interacted with AKR1C3 and potently inhibited the reduction of DAUN to C13-hydroxy-metabolite daunorubicinol (DAUN-OL) in a dose-dependent manner. Furthermore, Michaelis–Menten kinetics and the Lineweaver–Burk double reciprocal plot provided evidence of a mixed-type mode of inhibition (inhibition constant (Ki) = 2.9 μM, α > 1.0) (Figure 1B,C). 

### 3.2. Effect of ZAN on AKR1C3-Mediated DAUN Metabolism in Transfected HCT116 Cells

Next, we investigated the effect of ZAN on AKR1C3-mediated DAUN metabolism in intact cells. HCT116 cells were chosen because of their negligible endogenous expression of AKR1C3 and high transfectability [43]. HCT116 cells were transfected with pCI_AKR1C3 or an EV. Transfected cells were treated with DAUN (1 μM) and ZAN (or vehicle control), which were used at concentrations that were non-toxic to HCT116 cells, at a reaction interval of 2 and 4 h (data not shown). Figure 2 shows that ZAN, at all concentrations tested, significantly inhibited the conversion of DAUN to DAUN-OL in HCT116-AKR1C3 cells, proving that ZAN can influence the intracellular activity of the enzyme.

### 3.3. ZAN Overcomes DAUN Resistance in A549 and Transfected HCT116 Cells 

After confirming that ZAN inhibits AKR1C3 intracellularly, we analysed whether a decrease in DAUN metabolism modulates ANT resistance in HCT116 cells. HCT116-AKR1C3 and HCT116-EV cells were treated with various combinations of DAUN and ZAN. In HCT116 cells expressing AKR1C3, the value of IC_50_ for DAUN alone was almost two times higher than the value of IC_50_ obtained from experiments with HCT116-EV cells. These data demonstrate that the expression of AKR1C3 protects cancer cells from DAUN toxicity. Importantly, as can be seen from the shifts in IC_50_ values, the DAUN resistance of HCT116-AKR1C3 cells was reversed when ZAN (5 or 10 µM) was added to the culture (Figure 3A). No such significant changes in IC_50_ were found in experiments with HCT116-EV cells (Figure 3B), confirming the critical role of the ZAN-AKR1C3 interaction in the DAUN resistance reversal effect. Combination effects were precisely quantified using the Chou–Talalay method [47,48]. Combination index (CI) vs. fraction of cells affected (F_A_) plots were created. Cells that survived the treatment were considered unaffected. The values of CI for DAUN with ZAN (5 or 10 µM) resulted in synergism (CI = 0.3–0.9) in HCT116-AKR1C3 cells. The lowest value of CI was detected when 10 µM ZAN was combined with 0.25 µM DAUN (CI = 0.33 ± 0.11). In contrast, additivity to antagonism was detected when the examined drug combinations were tested in HCT116-EV cells (CI = 0.9–10) (Figure 3C,D). 

After confirming that the inhibition of AKR1C3 remains beyond the synergy between DAUN and ZAN in transfected cells, combination studies were performed with the lung adenocarcinoma cell line A549, which naturally expresses considerable amounts of AKR1C3 [43]. Similar to HCT116-AKR1C3 cells, significant shifts in IC_50_ values were detected when A549 cells were treated with a combination of DAUN and ZAN (1 and 5 μM) (Figure 4A). Using the Chou–Talalay method, a synergistic effect was detected over the entire range F_A,_ with the lowest CI value detected for the combination of 5 µM ZAN + 0.75 µM DAUN (CI = 0.06 ± 0.05) (Figure 4B).

### 3.4. ZAN Increases DAUN Accumulation by Inhibiting the Efflux Activity of ABC Transporters

As mentioned in the Introduction, not only AKR1C3, but also ABC transporters are IBR off-targets [41,49]. ABC transporters actively pump ANTs out of cancer cells, decreasing their effective concentration and contributing to drug resistance [50]. The fact that IBR re-sensitises drug-resistant ABCC1-overexpressing leukaemia cells to DOX [41] inspired us to investigate whether a similar interaction could be observed with ZAN. A549 and KG1α leukaemia cells were selected for our experiments because of their high ABC transporter expression. Using selective modulators LY335979 (inhibitor of ABCB1), Ko143 (ABCG2 inhibitor), and MK571 (ABCC1 inhibitor), along with substrate DAUN, we demonstrated the functional presence of the ABCB1 transporter in KG1α and the existence of ABCG2 and ABCC1 in the A549 cell line (Figure 5A,B). Furthermore, both the cell lines were incubated with a combination of DAUN and ZAN. The obtained data showed that ZAN at 10 and 25 μM inhibited DAUN efflux in A549 cells, while 5, 10, and 25 μM ZAN interacted with the ABCB1 transporter expressed in KG1α cells (Figure 5). Importantly, our results with model inhibitors correlate well with previous reports on the expressions and/or functional activities of examined transporters in selected cellular models [51,52,53,54]. 

### 3.5. ZAN Does Not Affect AKR1C3 Expression

In addition to inhibition, changes in AKR1C3 expression may affect the pharmacological fate of ANTs. Therefore, in the last part of our study, we investigated how ZAN affects the intracellular levels of AKR1C3 mRNA. Along with the KG1α leukaemia cells, the HepG2 hepatocarcinoma cell line was used as a liver model to evaluate whether ZAN has the potential to influence whole-body DAUN pharmacokinetics. First, we tested the effects of ZAN on the viability of HepG2 and KG1α cells to determine the concentration with tolerable cytotoxicity. Based on these experiments (Figure 6A,B), a pharmacologically relevant [55] and negligibly cytotoxic concentration of 0.5 μM ZAN was used for induction studies. The qRT-PCR results showed that ZAN did not cause any significant changes in AKR1C3 mRNA expression in either the KG1α or HepG2 cells (Figure 6C,D). These results confirmed that exposure to ZAN does not strengthen the resistance phenotype inside the cancer cells or force DAUN elimination from the body.

## 4. Discussion

The FDA approved ZAN as a second-line therapy for relapsed/refractory marginal zone lymphoma (MZL) and mantle cell lymphoma (MCL), and as a first-line treatment for Waldenström’s macroglobulinaemia [23,24,25]. Currently, ZAN, either alone or in combination with other drugs, is being studied in 66 clinical trials, of which 48 are currently recruiting, or have not started recruiting yet [56]. In addition to FDA-approved indications, the effectiveness of ZAN has been tested in other B-cell malignancies, such as CLL/small lymphocytic lymphoma (SLL), diffuse large B-cell lymphoma (DLBCL), CNS lymphoma, and follicular lymphoma [57,58,59,60,61]. CLL/SLL and indolent lymphomas are typically characterised by slow growth rates that allow a “watch and wait” strategy, if no symptoms are present [62,63,64,65]. However, genetic alterations leading to blastic transformations, such as Richter’s syndrome in B-CLL, can cause serious complications. They drive the aggression of cancer cells, and the disease progression requires immediate treatment [66,67]. Although rituximab + cyclophosphamide + ANT + vincristine + prednisone (R-CHOP) and other ANT-containing chemotherapy regimens are generally effective in such cases, some patients do not respond to the therapy. Patients who are naturally insensitive or resistant to drug therapy during treatment require new strategies to overcome resistance issues and prevent disease relapse [68].

Several factors that impair the effectiveness of ANT treatment in cancer cells have been identified. Metabolic inactivation and drug efflux, mediated by ABC transporters, are among the most important factors. AKR1C3 reduces DAUN and DOX to less potent C13-hydroxy-metabolites, whereas ABC transporters pump ANTs out of cancer cells, decreasing their effective concentration [30,31,32,33]. In our experiments, ZAN inhibited AKR1C3 at the level of a recombinant enzyme in AKR1C3-overexpressing intact cells. Synergism between DAUN and ZAN was detected in HCT116-AKR1C3 cells when ZAN was tested at concentrations of 5 and 10 µM. In A549 cells, a similar effect was observed with ZAN, even at a concentration of 1 µM, which can be explained by the dual inhibition of both AKR1C3 and ABC transporters. In a phase 1 study, a therapeutic dose of 320 mg resulted in a ZAN plasma Cmax of 1.4 µM [55]. Therefore, it is presumed that the effects observed in our in vitro experiments could also be manifested in patients with cancer. Importantly, no upregulation of AKR1C3 mRNA was detected upon ZAN treatment of HepG2 or KG1α cells, suggesting that ZAN does not affect systemic DAUN metabolism or influence the resistance of cancer cells to DAUN by promoting changes in AKR1C3 expression. 

There are several lines of evidence regarding the role of AKR1C3 and ABC transporters in aggressive lymphomas and their sensitivity to ANT therapy, identifying them as potential drug targets. First, AKR1C3 expression was detected in samples obtained from patients with DLBCL treated with anthracycline-based therapy. Second, the AKR1C3 single nucleotide polymorphism may influence the survival outcomes of patients receiving the CHOP regimen [69]. Finally, the role of ABC transporters in cancer responses to ANT-containing chemotherapy has been suggested in DLBCL [70], MCL [71], and Burkitt lymphoma [72]. In addition to the primary indication of BTK inhibitors in CLL and lymphomas, their efficacy has recently been demonstrated in AML and solid tumours [73,74,75,76]. Rushworth et al. (2014) discovered that BTK is phosphorylated (p-BTK), and is thus constitutively active in the majority of samples obtained from patients with AML. Importantly, ibrutinib was found to enhance the cytotoxic effect of DAUN in high p-BTK-expressing AML cells and in the U937 cell line, but not in CD34+ non-malignant cells [75]. However, the mechanism of DAUN-IBR synergism may potentially be BTK-independent [77]. Our current results suggest that inhibition of DAUN metabolism and/or its cellular efflux may help to explain the ANT-sensitising effects of the BTK inhibitor ZAN. Importantly, AKR1C3 is not only an ANT-reductase, but is also involved in the production of sex hormones and pro-proliferative prostaglandins [78,79,80]. The association between AKR1C3 expression and poor patient prognosis has been proven many times in different solid tumours (e.g., breast, prostate, and liver) [35,81,82,83]. Several studies have further pointed to a potential link between AKR1C3 and leukaemogenesis [78,80,84]. Moreover, maternal and offspring polymorphisms of AKR1C3 were found to be associated with an increased risk of the development of childhood leukaemia [85], contributing to the FDA’s list of AKR1C3 as a relevant target in paediatric oncology [86,87]. Therefore, by targeting AKR1C3, ZAN may affect both leukaemia- and hormone-dependent, as well as independent, cancers. This further supports the rationale for combining ZAN with ANT therapy in cancers with a high expression of AKR1C3 and/or ABC transporters. If confirmed in vivo, this information may be translated into an effective therapeutic strategy beneficial for a significant number of patients.

## Figures and Tables

**Figure 1 pharmaceutics-14-01994-f001:**
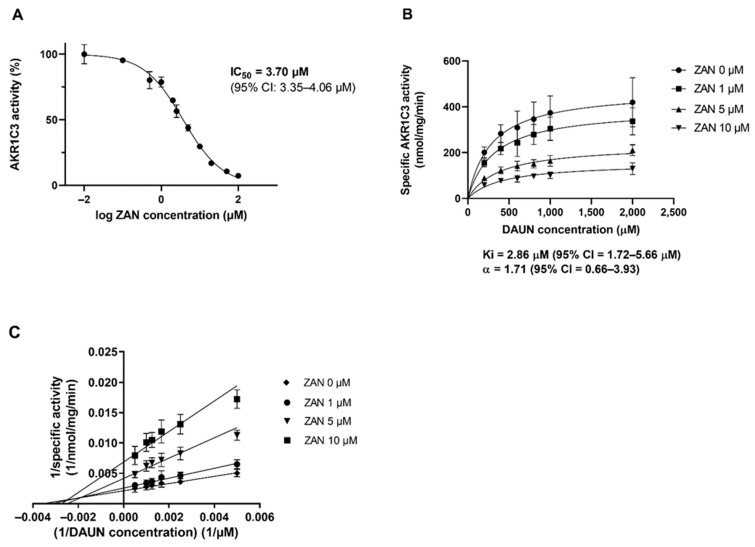
Zanubrutinib (ZAN) inhibits daunorubicin (DAUN) reduction catalysed by human recombinant AKR1C3. (**A**) IC_50_ of AKR1C3 inhibition. (**B**) Determination of the inhibition constant (Ki). (**C**) Mode of inhibition graphically represented by the Lineweaver–Burk plot. All data are expressed as the mean ± standard deviation (SD) from at least three independent experiments. 95% CI = 95% confidence interval.

**Figure 2 pharmaceutics-14-01994-f002:**
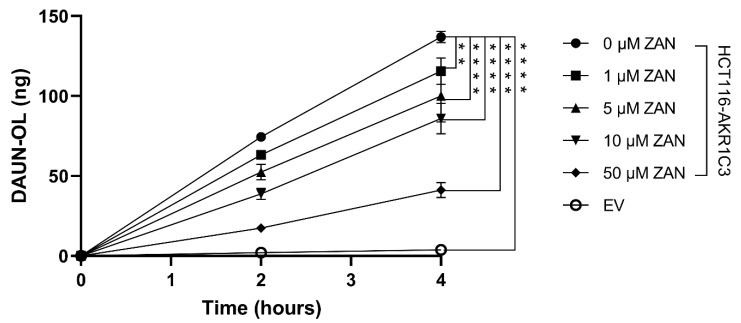
Zanubrutinib (ZAN) inhibits AKR1C3-mediated daunorubicin (DAUN) metabolism in intact cells. Transfected HCT116 cells were incubated with DAUN (1 µM) in combination with ZAN (1, 5, 10 and 50 µM) or its vehicle dimethyl sulfoxide (DMSO) for 2 and 4 h. DAUN-OL was then extracted and its amount was analysed by ultra-high-performance liquid chromatography (UHPLC). Data were evaluated with one-way analysis of variance (ANOVA), followed by Dunnett’s post hoc test. ** *p* < 0.01 and **** *p* < 0.0001 compared to DMSO vehicle control. The values are expressed as mean ± standard deviation (SD) from three independent experiments.

**Figure 3 pharmaceutics-14-01994-f003:**
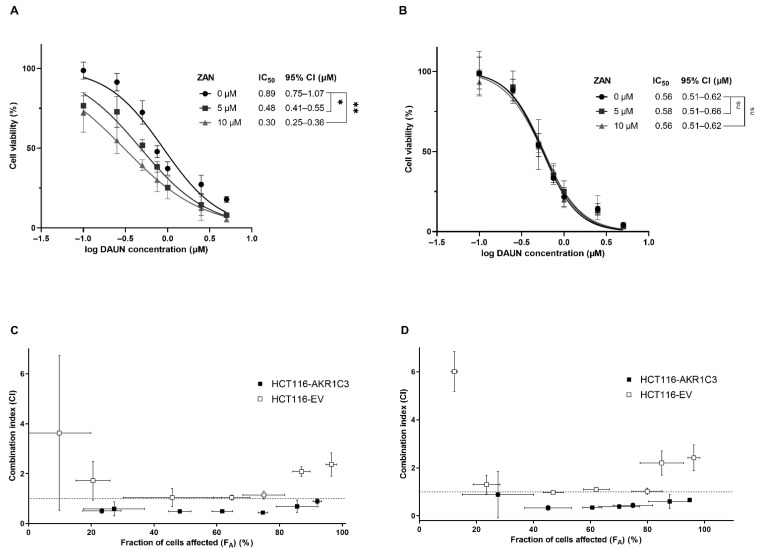
Zanubrutinib (ZAN) counteracts AKR1C3-mediated daunorubicin (DAUN) resistance in cancer cells. HCT116-AKR1C3 (**A**) and HCT116-EV (**B**) cells were treated with dimethyl sulfoxide (DMSO) (a vehicle) or ZAN (5 and 10 µM) and increasing concentrations of DAUN (0.1–5 µM). After 72 h of incubation, cell viability was evaluated using the XTT test. GraphPad Prism 9.3.1. was used for one-way analysis of variance (ANOVA), followed by Dunnett’s post hoc test. ns = non-significant, * *p* < 0.05, and ** *p* < 0.01 compared to DMSO vehicle control. 95% CI = 95% confidence interval. Chou–Talalay analysis was conducted to create combination index (CI) vs. fraction of cells affected (F_A_) plots for 5 (**C**) and 10 µM ZAN (**D**) in order to distinguish between synergism (<0.9), additivity (0.9–1.1), and antagonism (>1.1). The dotted line was positioned at CI = 1 for better orientation. Data are presented as mean ± standard deviation (SD) from three independent experiments.

**Figure 4 pharmaceutics-14-01994-f004:**
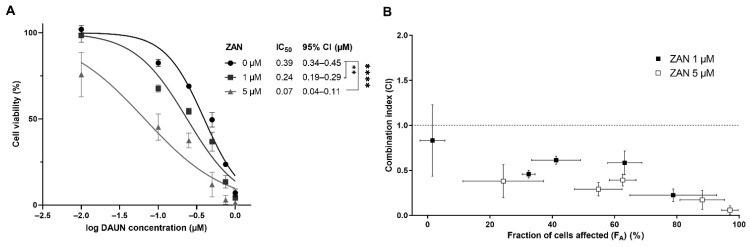
Zanubrutinib (ZAN) counteracts daunorubicin (DAUN) resistance in A549 cells. (**A**) A549 cells were treated with dimethyl sulfoxide (DMSO) (a vehicle) or ZAN (1 and 5 µM) and increasing concentrations of DAUN (0.01–1 µM), incubated for 72 h, and then cell viability was evaluated using the XTT test. GraphPad Prism 9.3.1. was used for one-way analysis of variance (ANOVA), followed by Dunnett’s post hoc test. ** *p* < 0.01 and **** *p* < 0.0001 in regards to DMSO vehicle control. 95% CI = 95% confidence interval. (**B**) Chou–Talalay analysis was performed to create the combination index (CI) vs. fraction of cells affected (F_A_) plot for 1 and 5 µM ZAN and to distinguish synergism (<0.9) from additivity (0.9–1.1) and antagonism (>1.1). The dotted line was positioned at CI = 1 for better orientation. Data are presented as mean ± standard deviation (SD) from three independent experiments.

**Figure 5 pharmaceutics-14-01994-f005:**
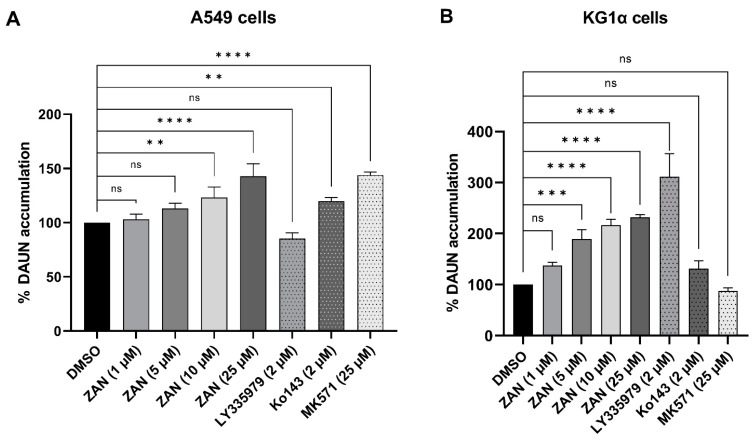
Flow cytometry analysis of daunorubicin (DAUN) accumulation in A549 (**A**) and KG1α cells (**B**). A549 and KG1α cells were preincubated with zanubrutinib (ZAN) (1, 5, 10, and 25 µM) or selective inhibitors of ABC transporters: LY335979 (inhibitor of ABCB1), Ko143 (inhibitor of ABCG2), or MK571 (inhibitor of ABCC1). After preincubation, 2 µM (A549) or 1 µM DAUN (KG1α) were added to the cells. The bar graphs represent the DAUN accumulation in % relative to the dimethyl sulfoxide (DMSO) vehicle control. Data were analysed by one-way analysis of variance (ANOVA), followed by Dunnett’s post hoc test; ns = non-significant, ** *p* < 0.01, *** *p* < 0.001, and **** *p* < 0.0001. Data are presented as mean ± standard deviation (SD) from three independent experiments.

**Figure 6 pharmaceutics-14-01994-f006:**
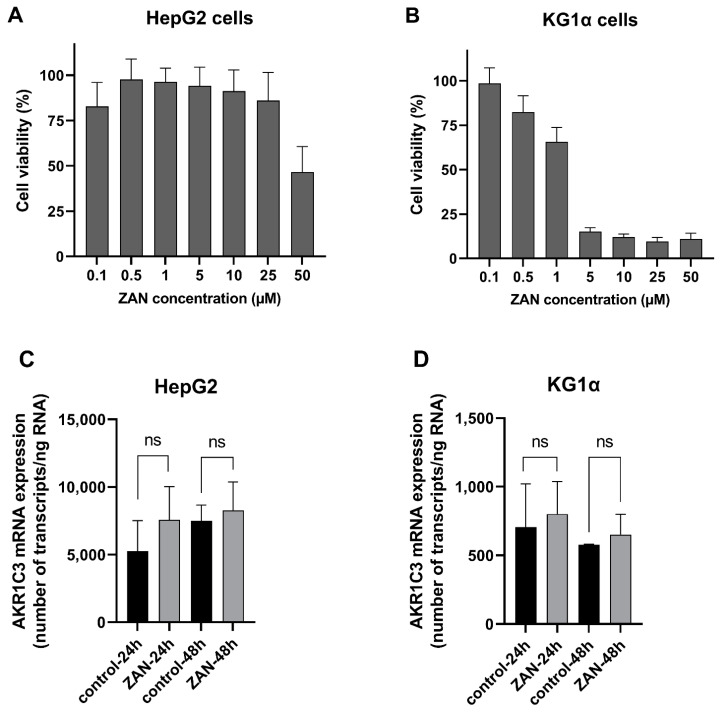
Effect of zanubrutinib (ZAN) on AKR1C3 mRNA expression in HepG2 and KG1α cell lines. To find a non-toxic ZAN concentration, HepG2 (**A**) and KG1α (**B**) cells were treated with ZAN (0.1–50 µM) or dimethyl sulfoxide (DMSO) (a vehicle) for 48 h, and the cell viability was evaluated by the XTT test. Furthermore, determination of AKR1C3 mRNA following ZAN (0.5 µM) exposure was performed in HepG2 (**C**) and KG1α (**D**) cells. qRT-PCR was used to monitor changes in AKR1C3 mRNA expression following a 24 and 48 h incubation. The Student’s unpaired t-test with Welch’s correction was used to assess statistical significance; ns = non-significant. The data represent mean ± standard deviation (SD) from at least three independent experiments.

## Data Availability

Not applicable.

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
