# Peer review of "Bruton’s Tyrosine Kinase Inhibitor Zanubrutinib Effectively Modulates Cancer Resistance by Inhibiting Anthracycline Metabolism and Efflux"

_pharmaceutics, 2022, doi:10.3390/pharmaceutics14101994_

Round 1

Reviewer 1 Report

A very interesting and well-designed study, which is impactful in the field of anthracyclines metabolism and search for novel AKR1C3 inhibitors.

I have some minor concerns:

1. There is a problem with fonts in the manuscript, which should be fixed. 

2. I suggest using "DOX" and "DAUN" abbreviations (instead of Dox and Daun) since other abbreviations are also uppercase. There is a wrong abbreviation on page 2, line 68 (Dau). Daun ol abbreviation has no explanation, as well as, Daun ol is not described as Daun metabolite. Page 2, line 88 it should be Fetal instead of Foetal.

3. 2.10 Induction studies in HepG2 and KG1a cells - cytotoxicity and induction studies should be described in separate paragraphs 

4. Fig. 3 C and D should be more legible. I guess the value of "1" is presented as the dotted line. This should be described in the caption. The "fraction of cells affected" parameter should be briefly described in one sentence (is it equal to cells viability?). In the text, the lowest value of CI should be pointed out.

5. I have doubts about mixed-type inhibition. Is Km increasing with increasing concentration of ZAN (fig. 1B)? Did the authors consider non-competitive inhibition?

6. I understand that results from experiments were compared to DMSO vehicle control but 0,5% DMSO concentration is quite high, especially for 72 h incubations. Did DMSO affect significantly cell viability in the highest concentrations or not? It should be declared in materials and methods. 

Author Response

A very interesting and well-designed study, which is impactful in the field of anthracyclines metabolism and search for novel AKR1C3 inhibitors.

I have some minor concerns:

  1. There is a problem with fonts in the manuscript, which should be fixed.

We corrected the fonts whenever we found this problem.

  1. I suggest using "DOX" and "DAUN" abbreviations (instead of Dox and Daun) since other abbreviations are also uppercase. There is a wrong abbreviation on page 2, line 68 (Dau). Daun ol abbreviation has no explanation, as well as, Daun ol is not described as Daun metabolite. Page 2, line 88 it should be Fetal instead of Foetal.

We corrected Dox, Daun and Daun-ol to DOX, DAUN and DAUN-OL in the text as well as in graphs (it is necessary to change all figures for the new ones in TIFF format). We explained that DAUN-OL is a C13-hydroxy-metabolite of DAUN. Also foetal was changed to fetal.

  1. 2.10 Induction studies in HepG2 and KG1a cells - cytotoxicity and induction studies should be described in separate paragraphs

We separated these methods into two paragraphs.

  1. Fig. 3 C and D should be more legible. I guess the value of "1" is presented as the dotted line. This should be described in the caption. The "fraction of cells affected" parameter should be briefly described in one sentence (is it equal to cells viability?). In the text, the lowest value of CI should be pointed out.

We explained the position of dotted line (lines 303 and 323) and fraction of cells affected (line 286). We added the information about the lowest CI value into manuscript (lines 289 and 312).

  1. I have doubts about mixed-type inhibition. Is Km increasing with increasing concentration of ZAN (fig. 1B)? Did the authors consider non-competitive inhibition?

In our results, the value of Km was increasing with increasing ZAN concentration except for 10 uM ZAN. Therefore, we used GraphPad equation for mixed model inhibition (https://www.graphpad.com/guides/prism/latest/curve-fitting/reg_mixered_model.htm). As stated in GraphPad prism: “The mixed model is a general equation that includes competitive, uncompetitive and noncompetitive inhibition as special cases. Alpha determines mechanism. Its value determines the degree to which the binding of inhibitor changes the affinity of the enzyme for substrate. Its value is always greater than zero. When Alpha=1, the inhibitor has equal affinity for the enzyme and the enzyme-substrate complex. This is identical to non-competitive inhibition. When Alpha>1, the inhibitor preferentially binds to the free enzyme. When Alpha is very large, binding is almost entirely to the free enzyme, and the mixed-model approaches competitive inhibition.

  1. I understand that results from experiments were compared to DMSO vehicle control but 0,5% DMSO concentration is quite high, especially for 72 h incubations. Did DMSO affect significantly cell viability in the highest concentrations or not? It should be declared in materials and methods.

The 0.5% DMSO concentration is a worldwide-accepted limit and is used in all our experimental works using cellular models. In fact, it is not unusual to use DMSO concentrations up to or over 1% in cellular experiments (see ref bellow). However, we agree that this is an important issue to be properly addressed. In our studies, next to using vehicle controls, each of the cellular model is screened for possible effect of DMSO on cells’ viabilities before experiments. We added the notice about this validation assays to the revised manuscript (the end of section 2.2)

Kuhlmann, M. K., E. Horsch, et al. (1998). "Reduction of cisplatin toxicity in cultured renal tubular cells by the bioflavonoid quercetin." Arch Toxicol 72(8): 536-40.

Danz, H., S. Stoyanova, et al. (2002). "Inhibitory activity of tryptanthrin on prostaglandin and leukotriene synthesis." Planta Med 68(10): 875-80.

Johnston, P. A. and P. A. Johnston (2002). "Cellular platforms for HTS: three case studies." Drug Discov Today 7(6): 353-63.

Lauth, M., A. Bergstrom, et al. (2007). "Inhibition of GLI-mediated transcription and tumor cell growth by small-molecule antagonists." Proc Natl Acad Sci U S A 104(20): 8455-60.

Reviewer 2 Report

The authors present a study on the synergistic effect between treatment with Zanubrutinib (Zan) and Daunorubicin (Daun) in different in vitro cancer models. The manuscript starts showing the inhibition of the catalytic activity of human recombinant AKR1C3 by ZAN. The inhibition of Daun metabolism by ZAN was verified using an AKR1C3 overexpressing colon cancer cell line. The decrease in cell viability induced by Daun upon ZAN treatment observed in the AKR1C3 overexpressing colon cancer cells, and a lung cancer cell line that naturally expresses high levels of AKR1C3, supports the synergism of the combination treatment. Considering that the activity of ABC transporters is an important mechanism for Daun resistance, the authors also report an increase in Daun accumulation induced by ZAN treatment. A similar effect can be observed when treating the cells with select ABC transporters inhibitors. Although the biochemical activity of AKR1C3 is affected by ZAN the authors could not observe any effect on the mRNA expression of AKR1C3 on a leukemia cell line and a liver cancer cell line.

The results on the accumulation of Daun upon ZAN treatment are promising. Still, information on the mRNA and protein levels of the ABC transporters would further strengthen the hypothesis suggested by the authors.

Because the authors show that the activity of the recombinant AKR1C3 is affected by ZAN, it would make sense that the effect is not related to transcriptional regulation. Even so, showing the mRNA and protein levels of AKR1C3 and the ABC transporters at their basal state, treated with Daun, and the combination with ZAN in all the cell models used in the manuscript would reinforce the authors' conclusions. 

It would enhance the quality and soundness of the manuscript if the data not shown could be included as supplementary information.

The manuscript is well written and clear.

Author Response

The authors present a study on the synergistic effect between treatment with Zanubrutinib (Zan) and Daunorubicin (Daun) in different in vitro cancer models. The manuscript starts showing the inhibition of the catalytic activity of human recombinant AKR1C3 by ZAN. The inhibition of Daun metabolism by ZAN was verified using an AKR1C3 overexpressing colon cancer cell line. The decrease in cell viability induced by Daun upon ZAN treatment observed in the AKR1C3 overexpressing colon cancer cells, and a lung cancer cell line that naturally expresses high levels of AKR1C3, supports the synergism of the combination treatment. Considering that the activity of ABC transporters is an important mechanism for Daun resistance, the authors also report an increase in Daun accumulation induced by ZAN treatment. A similar effect can be observed when treating the cells with select ABC transporters inhibitors. Although the biochemical activity of AKR1C3 is affected by ZAN the authors could not observe any effect on the mRNA expression of AKR1C3 on a leukemia cell line and a liver cancer cell line.

The results on the accumulation of Daun upon ZAN treatment are promising. Still, information on the mRNA and protein levels of the ABC transporters would further strengthen the hypothesis suggested by the authors.

Based on your suggestion, we performed a literature survey focusing on this issue. We found several articles reporting on the expression/functional activity of ABC transporters in selected cellular models. Importantly, the results correlate well with our data, thus confirming on the validity of our cells and experimental setup. Notice on this issue has been added to the end of section 3.4. (ref. 51-54 were added to the References)

Because the authors show that the activity of the recombinant AKR1C3 is affected by ZAN, it would make sense that the effect is not related to transcriptional regulation. Even so, showing the mRNA and protein levels of AKR1C3 and the ABC transporters at their basal state, treated with Daun, and the combination with ZAN in all the cell models used in the manuscript would reinforce the authors' conclusions.

Although we agree that such experiments might improve the quality of our manuscript, unfortunately, we are not able to perform them under minor revision status. Such wide set of experiments would take several weeks or even months to be conducted properly according to the common scientific rules. We believe, you can understand that we cannot follow your suggestion in this case.

It would enhance the quality and soundness of the manuscript if the data not shown could be included as supplementary information.

The term “data not shown” is mentioned in section 3.2. In this case, we measured effect of ZAN on the viability of transfected HCT116 cells. However, we obtain null effects in all variants, which is logical considering the extremely short incubation intervals (used in follow-up inhibitory assay). To avoid showing “empty message” results (i.e. graph with uniform 100% viability), we selected the “data not shown” type of presentation. Considering above mentioned details, we believe that this is the best option in this case and thus left the presentation type unchanged. Sure, if you stand on your suggestion, we can add the graph to the manuscript.

The manuscript is well written and clear.